# Visual Attention Patterns Toward Female Bodies in Anorexia Nervosa—An Eye-Tracking Study with Adolescents and Adults

**DOI:** 10.3390/bs15081027

**Published:** 2025-07-29

**Authors:** Valeska Stonawski, Oliver Kratz, Gunther H. Moll, Holmer Graap, Stefanie Horndasch

**Affiliations:** 1Department of Child and Adolescent Mental Health, University of Erlangen-Nuremberg, 91054 Erlangen, Germany; valeska.stonawski@uk-erlangen.de (V.S.); oliver.kratz@uk-erlangen.de (O.K.); gm@gunther-moll.de (G.H.M.); 2Department of Psychosomatic Medicine and Psychotherapy, University of Erlangen-Nuremberg, 91054 Erlangen, Germany; holmer.graap@uk-erlangen.de; 3Medical School, Department of Child and Adolescent Psychiatry and Psychotherapy, Bielefeld University, University Medical Center OWL, Protestant Hospital of Bethel Foundation, 33617 Bielefeld, Germany

**Keywords:** eye tracking, anorexia nervosa, adolescents, body stimuli

## Abstract

Attentional biases seem to play an important role in anorexia nervosa (AN). The objective of this study was to measure visual attention patterns toward female bodies in adolescents and adults with and without AN in order to explore developmental and disease-specific aspects. Female adult and adolescent patients with AN (n = 38) and control participants (n = 39) viewed standardized photographic stimuli showing women’s bodies from five BMI categories. The fixation times on the bodies and specific body parts were analyzed. Differences between participants with and without AN did not emerge: All participants showed increased attention toward the body, while adolescents displayed shorter fixation times on specific areas of the body than adults. Increased visual attention toward areas indicative of weight (e.g., hips, thighs, abdomen, buttocks) and a shorter fixation time on unclothed body parts were observed in all participants. There is evidence for the developmental effect of differential viewing patterns when looking at women’s bodies. The attention behavior of patients with AN seems to be similar to that of the control groups, which is partly consistent with, and partly contradictory to, previous studies.

## 1. Introduction

### 1.1. Cognitive and Attentional Biases in AN

Eating disorders (EDs) are thought to be characterized by dysfunctional cognitive schemas that influence information processing at multiple levels ([39]). These cognitions may bias attention toward ED-related stimuli, such that negative body schemas result in individuals attending to schema-consistent stimuli, potentially reinforcing a negative self-image ([42]). Especially for young people, appearance-related information is often conveyed visually, e.g., via images of (female) bodies on social media, and could significantly influence body dissatisfaction and ED symptomatology ([10]). The impact on and processing of such images in people with EDs are especially important as images are increasingly being used in therapeutic approaches, either via body exposure therapies ([6]), VR methods ([29]) or other imagery-based approaches ([35]). Two recent reviews state the importance of visual stimuli for those with ED pathology and the need to identify subtle information processing variations in EDs and their role in the development and maintenance of the disorders ([4]; [28]). One way to measure such more subtle attentional biases is by observing selective visual attention processes via eye tracking. Eye-tracking research has been widely used to provide insights into cognitive, social, and emotional processes in EDs via understanding the selection of visual information, which can be investigated at both early and later stages of attention in high temporal resolution. Further advantages of the eye-tracking method are that it allows for the continuous collection of data and requires fewer trials to robustly investigate processing strategies than the fMRI and EEG/ERP methods ([28]). As subjective ratings and objective responses towards emotionally salient stimuli often oppose each other, the use of objective methods alongside self-report techniques is highly relevant ([4]).

### 1.2. Attentional Biases Regarding Body Stimuli

In a review covering 14 studies on visual attention toward body stimuli, across ED diagnoses, there was evidence of attentional biases toward body stimuli ([21]). Specifically, patients with AN have been shown to pay more attention to body shapes—especially thin ones—when presented alongside images of social interactions ([27]). Another recent study examining the interplay of attention patterns toward pictures of bodies and social stimuli (human faces) using the same body stimulus set as the present study showed a bias toward bodies rather than faces in adolescents with AN, but also in a clinical, as well as a non-clinical, control group. However, this bias was most pronounced in adolescents with AN, particularly toward underweight bodies ([32]). In a very recent study, stronger attentional bias toward other women’s bodies and body parts in patients with AN than in HC (healthy controls) participants with a combined dot probe and eye-tracking methodology was shown ([24]). A review in non-clinical populations based on the findings of eye-tracking studies concluded that individuals with high levels of body dissatisfaction tend to orient more toward desired and feared appearance-related stimuli compared to control groups, with medium to large effect sizes ([30]). [8] ([8]) found that women with high body image avoidance tendencies viewed thin body stimuli (as opposed to overweight stimuli) longer than those with low avoidance behavior. This could be interpreted as an avoidance of aversive overweight stimuli; however, at the same time, this indicates an increased level of social comparison with the thin stimuli.

### 1.3. Attentional Biases Regarding Weight-Indicative Areas (WIA)

Certain areas of the body, such as the abdomen or hips, are a particularly common source of dissatisfaction among women. In contrast to other parts of the body, they are especially indicative of a person’s weight and often judged to be unattractive ([13]). Research findings suggest that increased visual attention is paid to these body regions when looking at others‘ bodies. A recent review suggests that biases with medium to large effect sizes are associated with the body regions that are typically used to evaluate weight changes in body-dissatisfied populations ([30]). For example, in an eye-tracking study, [40] ([40]) found that both participants with and without AN looked at the abdomen for the longest time of all body parts—with the exception of the head—in photographs of their own and other bodies. In our pilot study ([17]), we were able to show that the hips, thighs, abdomen, and buttocks were the areas viewed longer than the rest of the body by adolescent girls both with and without an ED. By contrast, an avoidance of body areas that are judged to be “problematic” and unattractive, as well as an avoidance of overweight body stimuli, has been observed in patients with AN ([41]). The results of another study ([19]) point to avoidant gaze behavior toward “problem areas” in women with a high drive for thinness and body dissatisfaction. Miquel-Nabau et al. investigated groups of non-clinical participants and found that those showing attentional biases towards weight-related body parts in a VR situation had a non-significant tendency towards higher body dissatisfaction and displayed a more balanced attentional pattern after Attentional Bias Modification training. Consequently, such potentially dysfunctional viewing patterns can be subject to modification ([25]).

Studies show that, when looking at their own bodies, participants with AN and bulimia nervosa (BN) pay more attention to parts of their body that they rate as unattractive compared to control participants, especially in earlier processing stages ([21]). For example, adolescents in general (with or without an ED) paid attention to unattractive body areas longer than to attractive areas; this effect was stronger in adolescents with AN, and was observed for both their own and others’ bodies ([2]). Similar behavior was shown in adults with an ED for their own, but not to others’, bodies ([23]). Hartmann et al. measured dwell times on body parts that are subjectively considered unattractive versus attractive, and the body parts that show weight status and weight gain the most strongly (stomach, hips, thighs) versus the least strongly when looking at one’s own body and a computer-generated obese body ([14]). They found increased attention to one’s own subjectively unattractive body parts as well as to body parts indicative of weight status or weight gain in patients with AN and control participants, but a stronger attentional bias (longer focus on subjectively unattractive versus attractive body parts) in women with AN was noted when confronted with an obese stimulus. [40] ([40]), by contrast, found in their study that participants with AN viewed their own thighs for a shorter period than the control subjects.

### 1.4. Attention Toward Unclothed Body Parts

In our pilot study, using black and white stimuli of females from different weight categories in underwear or swimwear, we found preferential attention to unclothed as opposed to clothed body parts, specifically in girls with EDs ([17]). The effect was most pronounced regarding normal-weight body stimuli. However, it remained unclear whether this phenomenon is present also in “pure” AN samples of adolescents and adults and whether it is linked more to clothing or more to weight-relatedness of body parts.

### 1.5. Aims and Hypotheses

In a sample of female adolescent and adult participants with AN and control participants (healthy controls = HC) of similar age, we were interested in visual attention allocation, measured via fixation times within certain regions of interest (ROIs), when confronted with stimuli of female bodies. The method of eye tracking has the advantage of measuring visual attention in an objective and direct way. Unlike indirect attentional tasks like the dot probe or emotional Stroop, which infer attention from reaction times, eye tracking provides high-resolution spatial and temporal data, making it more sensitive to subtle attentional biases.

Examining four participant groups (adolescents and adults, AN and HC groups each), we were able to explore developmental and disease-specific aspects at the same time. Building upon our pilot study ([17]), but using more standardized and naturalistic stimuli, we were able to differentiate between attention towards the whole body, the single body parts, and groups of body parts like WIA and unclothed regions. The aim was to gain a nuanced understanding of how distinct attentional phenomena might be present when looking at the same stimuli. To the authors’ knowledge, study results on attentional biases regarding single body parts—apart from the studies mentioned above—are still scarce.

**Hypothesis** **1** **(H1).**
*We hypothesized that there would be increased attention toward (a) the whole body (see e.g., [21]) and (b) unclothed body parts in patients with AN vs. HCs ([17]).*


**Hypothesis** **2** **(H2).**
*In line with previous studies, we hypothesized increased attention toward WIA in patients with AN and HCs (see [30]; [40]; [17]).*


In a more explorative way, without a strong base of previous research, we aimed to look at the differences between attentional patterns towards images of bodies of different weight categories and explore the developmental differences between adolescent and adult participants, comparing those age groups and looking at the interactions between groups and the bodies’ weight categories.

## 2. Materials and Methods

### 2.1. Sample

The sample consisted of a total of 77 female subjects, 38 of whom were diagnosed with AN, and 39 HCs. Of those, 42 participants were adolescents (age ≤ 18 years) and 35 were adults (>18 years). An a priori power analysis using GPower ([11]) indicated that a total sample size of 68 participants (n = 17 per group) was required to detect a medium effect size (f = 0.25) with 80% power (α = 0.05) in a mixed ANOVA with a 2 (participant Group: AN vs. HC) × 2 (age Group: adolescents vs. adults) between-subjects design and a 5-level within-subject factor (body weight category). The patients were undergoing treatment in the Department for Child and Adolescent Mental Health or the Department of Psychosomatic Medicine and Psychotherapy of the University Hospital Erlangen, and fulfilled the ICD-10 criteria for anorexia nervosa (ICD-10 F50.0), having been diagnosed by an experienced (child) psychiatrist or psychologist using the German version of the Clinical Assessment Scale for Child and Adolescent Psychopathology (CASCAP-D ([9])) for adolescents and the Structured Diagnostic Interview for Mental Disorders (DIPS) ([31]) for adults. All participants had normal or corrected-to-normal vision. Exclusion criteria for the control group were current, or a history of, psychiatric disorder (assessed by the abovementioned interviews); learning disabilities; obesity; and severe somatic symptoms, including neurological disease. Subjects in the control group were recruited through posters, flyers, and personal contacts.

Written informed consent was obtained from all participants and their parents (in the case of adolescents). This study was approved by the ethics committee of the University of Erlangen-Nuremberg and conducted in accordance with the Declaration of Helsinki.

### 2.2. Procedure

The experiment was part of a larger study investigating participants’ psychophysiological reactions toward disease-specific stimuli (other obtained data are published elsewhere and beyond the scope of this article; for example, for the EEG data, see [18] ([18])).

All participants completed the German version of the Eating Disorder Inventory-2 (EDI-2), which assesses ED psychopathology and comprises 91 items that are divided into 11 subscales ([36]).

For viewing the pictures, the participants were seated comfortably at a distance of 60 cm from a computer monitor (resolution 1024 × 768 pixels) and calibrated for eye movement recording. The room was dimly lit, and the participant’s head was placed lightly on a chin rest in order to control for head movement. Eye movements were registered during the viewing period by the Eyegaze Analysis System™ (Interactive Minds, Dresden, Germany), an infrared video-based binocular tracking system using the bright pupil method with a temporal resolution of 60 Hz.

The body images appeared for 8 s each, with a one-second interstimulus interval. After the picture presentation, the participants were instructed to give subjective assessments of the body weight of the depicted women and their attractiveness by means of a nine-point visual analog scale. For the results of the weight and attractiveness ratings in a widely overlapping sample, see [16] ([16]).

### 2.3. Stimuli

Standardized greyscale photographic stimuli showing women’s bodies in underwear in four positions (front, rear, profile standing, and profile sitting) were used (see [16]). Eight pictures from each BMI category were shown to all participants, which were categorized as follows: extremely underweight (BMI 13.5–15.0 kg/m^2^), underweight (BMI 17.0–18.0 kg/m^2^), normal weight (BMI 20.0–22.5 kg/m^2^), overweight (BMI 36.0–38.0 kg/m^2^), and extremely overweight (BMI > 50 kg/m^2^). The examples are presented in Figure 1.

### 2.4. Data Acquisition and Analysis

From the original 89 subjects, 12 subjects were excluded from the calculations due to technical problems or poor data quality. Data were included if a minimum total fixation time of 5000 ms/picture was ensured in at least 50% of the pictures in each category.

According to the hypotheses, the area occupied by the whole body, the WIA (which includes the hips, thighs, abdomen, and buttocks), and the unclothed areas of each body, respectively, were marked as regions of interest (ROI). A fixation was counted if the participants’ gaze was directed to a predefined area of one degree for at least 100 ms ([38]). The total fixation time within each ROI was calculated and corrected for the size of the body (parts) relative to the whole picture.

For the effects of age, BMI, and eating disorder symptoms (EDI-2 total and body dissatisfaction scores) between participant groups (patients with AN vs. HC) and age groups, *t*-tests were computed. For the gaze parameter “total fixation time within the ROIs relative to area of the ROIs and relative to total fixation time within the picture”, ANOVAs were conducted with the weight category of the depicted body factor (extremely underweight, underweight, normal weight, overweight, or extremely overweight) and the group factors participant group (patients with AN vs. HC), and age group (adolescents vs. adults). Separate ANOVAs were performed for the “total body”, “WIA”, and “unclothed body parts” ROIs. For “WIA” and “unclothed body parts”, another categorial factor region (WIA/unclothed body parts vs. the rest of the body) was included in order to control for the effects of long gaze durations outside the body area. A Greenhouse–Geisser correction was used for the ANOVAs. For specific group comparisons (patients with AN vs. HCs; adults vs. adolescents), post hoc *t*-tests were computed when the group effects or category×group interactions were shown. For significant category effects, post hoc Bonferroni-adjusted pairwise comparisons were performed. When significant group effects were seen, Pearson correlations, with eating disorder symptoms measured via the EDI and the EDI body dissatisfaction subscale, were calculated.

## 3. Results

### 3.1. Participant Characteristics

The mean age of the adult subjects was 26.8 years (standard deviation, SD = 7.3), and that of the adolescents was 15.7 years (SD = 2.0). For the participant characteristics in the two groups, see Table 1. Significant differences emerged regarding psychopathology, which was measured via the EDI-2 total score and the body dissatisfaction subscale, between the adolescent and adult subgroups (total score: t(65) = −2.80, *p* = 0.007, d = −0.69; body dissatisfaction subscale: t(76) = −2.12, *p* = 0.038, d = −0.48), with higher values for the adult than adolescent participants. The adolescent and adult control participants did not differ regarding those values (|t| < 0.9, n.s.), whereas the adults and adolescents with AN differed (total score: t(28) = −3.84, *p* < 0.001, d = −1.48; body dissatisfaction subscale: t(36) = −2.29, *p* = 0.028, d = −0.74); this effect seems to be mainly driven by the AN group.

One or more comorbid psychiatric disorders were present in 15 adult participants (12: depressive episode; 1: anxiety disorder; 4: obsessive–compulsive disorder; 1: post-traumatic stress disorder) and 17 adolescents (16: depressive episode; 3: anxiety disorder; 2: obsessive–compulsive disorder; 1: personality disorder).

### 3.2. Visual Attention Patterns

#### 3.2.1. Total Body Area

In the ANOVA, a significant category effect emerged (see Table 2 for the ANOVA results). A post hoc Bonferroni-adjusted pairwise comparison between the individual categories regarding the fixation time in the whole body area yielded similar durations for the underweight and normal-weight categories but significant differences for all other categories. The participants looked at extremely underweight bodies for the longest time, followed by underweight/normal weight, overweight, and extremely overweight (shortest relative fixation time) bodies (see Appendix A).

In addition, a significant category × age group interaction effect was observed. Furthermore, *t*-tests examining the differences in viewing duration between adolescents and adults for each category revealed significant longer durations in adults only with respect to the lower two weight categories, extremely underweight and underweight, and a trend toward significance was also noted for the normal-weight category (see Figure 2 and Appendix A).

Regarding the age group effect, the Pearson correlations with eating disorder symptoms, measured via the EDI-2 total score and body dissatisfaction subscale, were calculated for the adolescents and adults subgroups, as the two groups differed regarding those values. Within both groups, no significant correlations between the total score or body dissatisfaction subscale and relative fixation times for the categories of extremely underweight up to normal weight were found (all *p* ≥ 0.75, all r ≤ |0.315|).

#### 3.2.2. Unclothed vs. Clothed Body Parts

When analyzing the fixation time on unclothed vs. clothed body parts, a significant main effect of region indicated that unclothed body parts were fixated on for a significantly shorter time than the clothed regions of the body (see Table 2). Further, a main category effect was seen. A post hoc Bonferroni-adjusted pairwise comparison between the single categories yielded similar durations for the extremely underweight, underweight, and normal-weight categories, but significant differences were found between all other categories (see Appendix A). Therefore, the longest fixation time on unclothed body parts occurred in all participants when viewing the lowest three weight categories compared to the higher weight categories (see Figure 3). No age or participant group effects were seen.

#### 3.2.3. WIA

Regarding the ANOVA analyzing fixation time on WIA as opposed to the rest of the body, the main effect of region indicated a longer fixation time on WIA than on the rest in all participants. In addition, the main effect of the category emerged with a longer fixation time on WIA in extremely underweight and normal-weight pictures, followed by the underweight and two overweight categories (see Table 2 for the ANOVA results and Appendix A for the pairwise comparisons).

## 4. Discussion

In our sample of adolescent and adult participants with AN and age-matched control participants, we investigated attentional patterns via eye tracking when presented with images of female bodies.

Regarding ED symptomatology, differences between adult and adolescent participants emerged: adults, in general, showed higher ED symptom scores and a greater extent of body dissatisfaction. However, this was mainly driven by the fact that the adult patients with AN reported higher scores than the adolescent patients, hinting toward a more chronic and severe course of the disease. Our control groups of healthy adolescent girls and adult women reported similar scores (total score and body dissatisfaction) to the adult and adolescent normative populations, respectively ([20]; [36]).

Regarding the patterns of visual attention, many previous results could not be replicated. Our hypothesis, H1a, suggesting increased attention toward the whole body in patients with AN compared to the control participants, could not be confirmed. Of note, in the task used in this study—in contrast to other studies, such as [27]; [32]—there were no other stimuli competing for attention, so making a comparison between studies is difficult due to their different methodologies. However, all participants showed increased attention toward the body for the extremely underweight to normal-weight categories compared to the two overweight categories. This could be explained by the fact that the body’s area—which is naturally larger in more overweight bodies—was taken into account for the calculations, and therefore produced a “floor effect” for relative fixation times (with the generally low values within this category due to division through large area values). Therefore, results like those of young women with high body dissatisfaction showing sustained attentional biases to thin and also fat body images in comparison to more neutral stimuli could not be confirmed ([12]). Another previous study has shown that participants with BN looked at bodies with a lower BMI for a significantly longer time, and for significantly shorter times at high BMI bodies than the control participants ([3]). A similar pattern has been observed in university students, reporting a high use of avoidance strategies ([8]). This was true for all participants in our study, potentially indicating an avoidance of body shapes that are perceived as aversive by all participants; see the previous results for an overlapping sample ([16]). Therefore, anxiety when confronted with those (extremely) overweight bodies, which could, in turn, cause attentional avoidance, could play a role ([26]). Using the same stimulus set, regarding event-related potential measured via EEG ([18]), an increased late positive potential could be observed in patients with AN when confronted with extremely underweight body pictures. The central nervous mechanisms that possibly reflect highly automatic “motivated attention”, therefore, do not seem directly related to visual attention. Differences in attentional biases between patients with AN and healthy controls might not have emerged due to variability in individual symptom severity or compensatory attentional strategies (e.g., deliberate avoidance of salient body areas; for effects when looking at one’s own body, see [33]), which could obscure the group differences in gaze patterns.

We hypothesized that patients with AN display increased visual attention towards unclothed body parts (H1b). However, both patients and control participants showed shorter fixation times towards unclothed compared to clothed body parts. Since the clothed areas were smaller and the fixation times were related to the ROI area, this might be a statistical artifact, especially in higher stimulus weight categories. Different explanations for attentional avoidance of unclothed body parts could be that they are perceived as “threatening” in the context of social comparison or that directing the gaze to clothed rather than unclothed areas may be a strategic, socially guided behavior in the context of cultural or moral expectations.

Further, we hypothesized an increased visual attention toward areas indicative of weight (H2), which are often the most disliked body parts in patients with AN, but (similar to our pilot results) also in participants without an ED. This was confirmed, as all participants showed increased attention toward WIA compared to the rest of the body. A study involving participants without a psychiatric disorder showed that those with a high drive for thinness, especially females, focus more on the waist, hips, legs, and arms, and less on the face of whole body stimuli ([15]). This pattern was found in our ED sample and control group, though a selection bias could be present, as individuals who are interested in body image might be more likely to participate.

In participants with obesity and normal weight, Leehr et al. examined gaze behavior regarding the visual stimuli of obese bodies and found no differences between the groups; all participants looked longer at the waist of obese bodies and at the head of normal-weight bodies ([22]). In our study, the WIA stimuli were mainly looked at in extremely underweight and normal-weight bodies, probably due to the described “floor effect”.

Differences regarding attention patterns toward bodies in different weight categories could not be found without any interaction between the group and stimulus categories. This similar pattern in both groups differed; however, with regard to the ROIs examined for the whole body, all participants showed the longest fixation times for extremely underweight pictures, possibly because these bodies are novel stimuli, thus eliciting stronger emotions. For WIA, extremely underweight and normal-weight categories yielded similar fixation times, and for unclothed body parts, all three of the lowest categories were viewed similarly.

We examined the developmental effects and found that adolescents with and without AN had shorter fixation times within the whole body area shown in pictures compared to adults. This difference was significant for extremely underweight, underweight, and normal-weight pictures. One reason could be that the photos depicted adult women, who are more suitable for comparison by adults than adolescents. Adolescents may compare their appearance more with that of their peers (see e.g., [1]; [5]). Additionally, adults might have internalized beauty ideals more strongly, which could result in higher eating disorder symptomatology and body dissatisfaction. Visual avoidance of overweight bodies and body parts, investigated throughout a sample of adolescent and adult patients with AN, occurred at earlier processing stages (500 ms after stimulus presentation) in older patients and those with longer illness durations. A similar phenomenon could also have contributed to the described effect regarding lower weight categories and hints toward influences of age and chronicity, which should be explored further ([24]).

This aligns with findings that body satisfaction influences selective visual attention to thin bodies. Exposure to thin body images can cause average-sized bodies to appear larger, mediated by body size and shape misperception ([7]). Individuals who were less satisfied with their bodies showed a higher number and duration of fixations on thin bodies ([34]). However, no correlations were found between the EDI-2 total scores or body dissatisfaction subscales within age groups, suggesting that these phenomena are not directly related.

Further limitations of this study are its small sample size, the high rate of drop-outs due to poor data quality, and especially the fact that only a few patients underwent two measurements. Further comparability to other studies is only partly possible because of the different methodologies used (e.g., eye-tracking apparatus, stimuli, and the time period of their presentation). It also would have been interesting to investigate additional eye-tracking parameters like avoidance processes (via measuring fixations, e.g., on distraction stimuli), the first fixations, the number of fixations, and revisits to elucidate further aspects of attention. Likewise, the eye-tracking methodology enables us to add pupil diameter as a measure of emotional reactions towards stimuli to the setup. This is planned for future studies. Also, the sample only consists of severely ill inpatients and female patients, which may make it less representative of all patients with AN. As the stimuli depict adult women, they might be more appropriate for our adult participants than for our adolescent participants. Other factors that could affect visual attention, such as psychiatric comorbidities or individual differences in body image concerns, should be more closely investigated in further studies.

The strengths of the current investigation are the highly standardized but naturalistic picture material and the inclusion of two age groups to highlight the developmental course of attentional processes, phenomena that are highly subject to developmental influences. The results of this study contribute to a larger picture of attentional processes in EDs, pointing the way to further research, e.g., regarding developments across the life span or interventions targeting those subtle phenomena.

## 5. Conclusions

In our study regarding examining visual attention via eye tracking in patients with AN when presented with pictures of female bodies, we found differential patterns depending on what female body shape was shown in the photographs, and—regarding one partial hypothesis—depending on the participants’ age. For future research, it would be interesting to examine the effects of short-term interventions, such as tasks involving attention modification ([23]) or interventions targeting body image ([37]), which have been shown to modify visual attentional biases. Attentional Bias Modification training, for example, has been shown to reduce attentional bias to weight-related body parts ([25]). It would also make sense to include (by modification within the task or simply by controlling for such phenomena via subjective reports) aspects of social comparison and avoidance, and examine their role in attention allocation and bias ([30]).

## Figures and Tables

**Figure 1 behavsci-15-01027-f001:**
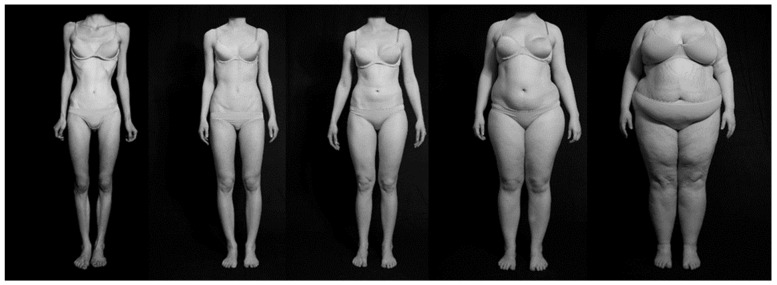
Photographic stimuli showing women’s bodies from five BMI categories (from left to right): strongly underweight, underweight, normal weight, overweight, and strongly overweight.

**Figure 2 behavsci-15-01027-f002:**
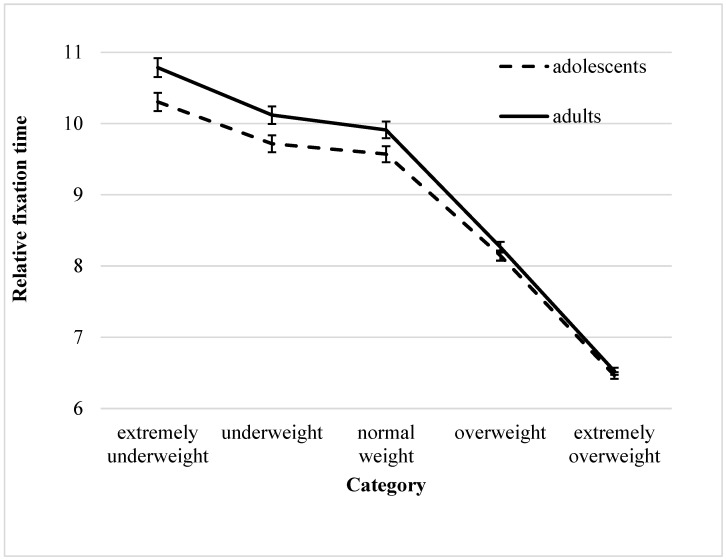
Fixation times on the total body relative to area and total fixation time (ms/ms) in the photographic stimuli of adolescents and adults (mean ± 1 SE).

**Figure 3 behavsci-15-01027-f003:**
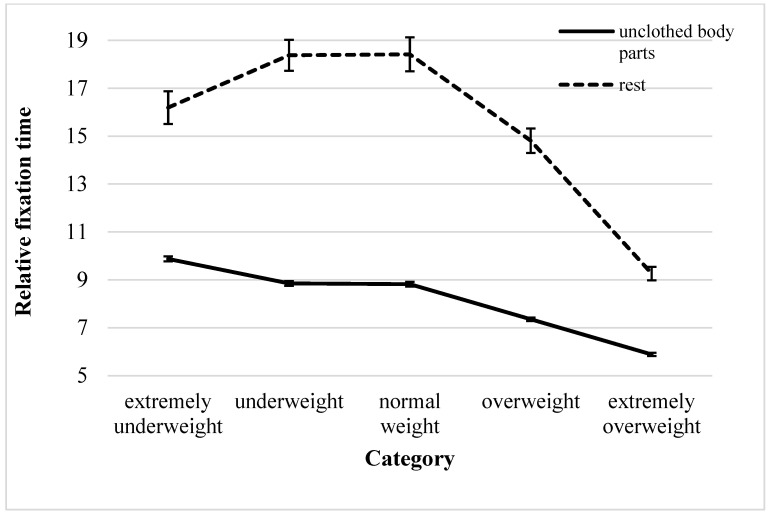
Fixation times within the unclothed and clothed body parts relative to the area and total fixation time (ms/ms) within the picture for the total sample (mean ± 1 SE).

**Table 1 behavsci-15-01027-t001:** Participant characteristics (means and SDs are reported).

Age Group	Adolescents (n = 42)			Adults (n = 35)		
Participant Group	AN (n = 20)	HC (n = 22)	*p*	AN (n = 18)	HC (n = 17)	*p*
Demographic characteristics						
Age (years)	15.52 (1.71)	16.12 (2.25)	n.s.	26.40 (7.08)	27.46 (7.68)	n.s.
BMI (kg/m^2^)	16.30 (1.59)	20.71 (3.21)	<0.001	16.12 (1.84)	21.44 (1.64)	<0.001
BMI age percentile	6.81 (10.11)	47.84 (26.49)	<0.001			
Clinical characteristics						
EDI-2 total score	274.65 (71.44)	207.82 (46.56)	0.001	349.94 (34.94)	216.88 (48.16)	<0.001
EDI-2 body dissatisfaction	35.79 (11.89)	26.04 (9.34)	0.003	42.36 (8.07)	29.70 (9.36)	<0.001

n.s. = not significant.

**Table 2 behavsci-15-01027-t002:** Results of the ANOVAs regarding the whole body area, WIA, and unclothed body parts (measurement t1).

Factor/Interaction	df	Error df	F	Partial η^2^	*p*
**whole body**					
category	4	70	1109.32	0.94	<0.001
category × age group	4	70	3.46	0.05	0.015
category × participant group	4	70	0.54	0.01	n.s.
age group	1	70	5.97	0.08	0.017
participant group	1	70	0.00	0.00	n.s.
age group × participant group	1	70	0.98	0.01	n.s.
**unclothed body parts**					
region	1	72	274.10	0.79	<0.001
category	4	69	143.76	0.67	<0.001
category × age group	4	69	0.80	0.01	n.s.
category × participant group	4	69	1.82	0.03	n.s.
age group	1	72	2.51	0.03	n.s.
participant group	1	72	1.43	0.02	n.s.
age group × participant group	1	72	1.94	0.03	n.s.
**WIA**					
region	1	71	100.95	0.59	<0.001
category	4	68	598.10	0.89	<0.001
category × age group	4	68	0.89	0.01	n.s.
category × participant group	4	68	1.54	0.02	n.s.
age group	1	71	0.97	0.01	n.s.
participant group	1	71	1.27	0.02	n.s.
age group × participant group	1	71	0.12	0.00	n.s.

n.s. = not significant.

## Data Availability

The data presented in this study are available on request from the corresponding author due to privacy reasons.

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
