# Peer review of "Visual Attention Patterns Toward Female Bodies in Anorexia Nervosa—An Eye-Tracking Study with Adolescents and Adults"

_behavsci, 2025, doi:10.3390/bs15081027_

Round 1

Reviewer 1 Report

Comments and Suggestions for Authors

General Comment

This study examines the visual attentional patterns towards female bodies of different weight categories amongst adults and adolescents with AN using an eye tracker. Participants were grouped according to age group (adults vs adolescents) and diagnostic status based on diagnostic interviews (AN vs health controls). The within-subjects variables included weight categories of body stimuli shown, various areas of the body, clothed vs unclothed body parts, and time of measurement (beginning vs end of treatment).

First of all, I would like to commend the authors on their impressive data collection which can make important contributions to the existing literature and have helpful clinical implications. However, my impression reading this manuscript is that the authors were overly ambitious in what they tried to include, and the number of variables examined. As a result, the manuscript read like a collection of semi-related investigations put together without a coherent linking narrative, as opposed to having a strong rationale based on theoretical/empirical background to start with that directly led to clear concise research questions and hypotheses, which then followed on to clinical or empirical implications. For example, the investigation of attention towards unclothed body parts and the effect of treatment on attentional patterns, in particular, seemed like “afterthoughts” or “add-ons” to the study, both in the brief and abrupt way that it was presented in the Introduction and cursory way it was mentioned in the rest of the manuscript. Specifically, the rationale for and clinical implication of studying attention towards unclothed body parts was unclear. Moreover, the question around the effect of treatment on attentional biases is arguably the most clinically interesting to readers, but the authors did not do it justice as the treatment section in the Introduction was very brief and lacked an in-depth discussion of the literature, rationale for the current investigation, and other details (e.g., what does an ABM protocol actually involve? Was that protocol included in this study?). See also comment #8. The findings were also not followed by an elaborated discussion. Another unfortunate consequence of the vast number of variables examined in this study is that the Results section was very hard to follow, simply due to the sheer density of the data presented in all the tables, figures, and text. If this was a PhD dissertation, there might have been scope for describing a multi-stage research program covering all of these various investigations whilst maintaining a coherent research narrative. However, as a condensed research publication, this brief examination of a vast number of variables ultimately “diluted” or detracted the manuscript from having a solid focus and compelling argument. Overall, I think the manuscript can be strengthened by concentrating on a few key complementary investigations that either add something to the existing literature, or address important theoretical and/or clinical questions, followed by a concise presentation of results and an elaborated discussion of interpretation and implications.

Some other comments:

  1. The Abstract would benefit from being more specific. E.g., line 21 – giving examples of “areas indicative of weight”, line 22 – which subsample?
  2. The more sensitive way to refer to the clinical population is “patients with AN” as opposed to “AN patients” (line 44). The former term was used in other areas of the manuscript, it is best to keep it consistent.
  3. “Stronger attentional bias” (line 99) can refer to both increased attention to or avoidance of stimuli, since both are considered biases in attentional processes. Therefore, it is better to be specific regarding the direction of the bias.
  4. Lines 109-110, the authors should be specific about how “visual attention allocation” was operationalised in this study.
  5. The statement regarding the advantage of using eye tracking in lines 110-112 requires more elaborate discussion and reasoning, as it could be disputed by some that it is the most “direct” way of measuring attention. Also, what are some of the “other attentional tasks” mentioned?
  6. The aim of the exploratory investigations (lines 122-123) was a bit unclear – were the authors interested in looking at the differences in attentional patterns towards images of bodies of different weight categories between groups, or across the groups, or an interaction?
  7. Why were individuals who were obese or “severe somatic” excluded from the study?
  8. The authors need to provide much more detail about the treatment offered in the Method section (lines 171-172), did it involve an ABM protocol since it was mentioned in the Introduction? What exactly did the treatment protocol involve? If the authors want to draw conclusions about the impact of treatment, then the treatment administered needs to be outlined in detail. This point was also raised in the initial main comment.
  9. The authors found that there was a significant difference in EDI-2 total score and body dissatisfaction subscale between the adult and adolescent subgroups. Although there was no significant correlation between those symptom scores and fixation time, it would have been more statistically sound for those scores to be included as a covariate in all analyses involving age group comparisons in order to control for the effects of this factor.
  10. Related to the point already mentioned in the initial main comment, the Results was very hard to follow particularly because multiple tables were often stacked and referred to in the same paragraph. This required a lot of flipping back and forth, looking at the dense tables and then searching for the relevant descriptive text to make sense of the findings. The table labels (S1, S2, etc) were also confusing.
  11. The discussion about the improvement of body dissatisfaction observed between ages 16-24 found in previous study not being replicated in this study (lines 317-319) seemed out-of-place, as it was not discussed or mentioned anywhere else in the manuscript as a point of interest.
  12. Could the finding that all participants showed shorter fixation time towards unclothed vs clothed body parts also simply be related to some sort of social desirability effect?
  13. Lines 415-418 seem to be the only place in the manuscript where the strength or rationale for the study is mentioned, which comes too late in the piece. Related to the main comment, the manuscript lacks a strong rationale from the beginning through to the end about how the study adds to the existing literature and why it is important.
  14. Various typos were found throughout the manuscript, e.g., line 148 “reactios”, line 156 “dimly-lit” instead of “dimly lighted”, line 291 “und”, line 390 “patters”, etc.
Comments on the Quality of English Language

The manuscript requires more thorough editing and proof-reading to rectify grammatical errors and typos, and improve clarity of expression. 

Author Response

General response:

Dear Reviewer,

Thank you for your helpful comments, especially on your general impression of the paper.

We understand that the multitude of research questions and their discussion might seem not structured and concise enough. Especially unclear seemed the relevance and meaning of the effects which occurred post- vs pre-treatment, given that on the one hand the sample performing the experiment at the two time points is very small and on the other hand the nature of the treatment which the patient underwent seems not adequately described. The multimodal inpatient treatment did not explicitly involve an ABM protocol; ABM as a treatment option was solely discussed as a possible therapeutic approach in the introduction. Other reviewers also mentioned those limitations to the research question of the longitudinal course of visual attention phenomena. By contrast, they did not have the impression that the research questions at time point 1 involving different regions of interest (whole body/WIA/clothed vs. unclothed) were too broad.  

Therefore we

  • deleted the passages involving the pre- vs. post-treatment effects and the respective table/figure and discussion which makes the paper shorter and – hopefully – more clear
  • explained more the rationale for performing several sub-analyses with the different ROIs. The aspect of clothed vs. unclothed body parts is indeed the research question which stems from our pilot results in adolescents with EDs and which we aimed to confirm or reject with the help of the current, more standardised stimuli. We explicitly wanted to see whether potential phenomena of visual attention are present also in “pure” AN samples of adolescents and adults, whether they are related to age effects and whether they are linked more to clothing or more to weight-relatedness of body parts. We added a sentence for clarification to the introduction (ll.96-99).

Reply to other comments:

  • The Abstract would benefit from being more specific. E.g., line 21 – giving examples of “areas indicative of weight”, line 22 – which subsample?

We clarified the “areas indicative of weight” in the abstract (l. 21). The subsample is now no longer mentioned (see General response). 

  • The more sensitive way to refer to the clinical population is “patients with AN” as opposed to “AN patients” (line 44). The former term was used in other areas of the manuscript, it is best to keep it consistent.

Thank you very much for pointing this out; we edited this throughout the manuscript in order to maintain a more sensitive language use.  

  • “Stronger attentional bias” (line 99) can refer to both increased attention to or avoidance of stimuli, since both are considered biases in attentional processes. Therefore, it is better to be specific regarding the direction of the bias.

In that case AN patients were focusing longer on subjectively unattractive versus attractive body parts; we clarified this in the text (l. 88).

  • Lines 109-110, the authors should be specific about how “visual attention allocation” was operationalised in this study.

We added an explanation accordingly.

  • The statement regarding the advantage of using eye tracking in lines 110-112 requires more elaborate discussion and reasoning, as it could be disputed by some that it is the most “direct” way of measuring attention. Also, what are some of the “other attentional tasks” mentioned?

We agree with this comment and elaborated a bit more on the characteristics of the measurement of attention via eye tracking, including naming examples of other, indirect tasks (ll. 107-109).

  • The aim of the exploratory investigations (lines 122-123) was a bit unclear – were the authors interested in looking at the differences in attentional patterns towards images of bodies of different weight categories between groups, or across the groups, or an interaction?

Thank you for this valuable comment; we aimed to clarify this point by adding that both – comparisons of groups and interactions with weight categories – should be explored, like specified in the methods section (ll. 125/126).

  • Why were individuals who were obese or “severe somatic” excluded from the study?

Girls and women with obesity were excluded, because attentional biases have been shown in obesity via eye tracking (e.g. Baur J, Krohmer K, Naumann E, Svaldi J. Attentional processing of body images in women with overweight and obesity. Eat Weight Disord. 2022 Oct;27(7):2811-2819. doi: 10.1007/s40519-022-01419-1. Epub 2022 Jul 4. PMID: 35781634; PMCID: PMC9556367). Those effects would have been required to consider separately, e.g. via two control groups (obese and normal weight).

Sorry, this expression “severe somatic, including neurological disease” is indeed a bit unclear. The aim of excluding participants with severe neurological disease (actually, no participant had to be excluded for those reasons) should minimize the risk that eye movements and/or the ability to focus are impaired by other factors not related to psychopathology. We shortened this to “severe neurological disease” (ll. 140/141) – as mentioned, this criterion didn’t have to be applied for any participant.  

  • The authors need to provide much more detail about the treatment offered in the Method section (lines 171-172), did it involve an ABM protocol since it was mentioned in the Introduction? What exactly did the treatment protocol involve? If the authors want to draw conclusions about the impact of treatment, then the treatment administered needs to be outlined in detail. This point was also raised in the initial main comment.

We totally agree with this point and – as mentioned in the General comments above – therefore decided to not consider the treatment and its effect on AB in the current version of the paper any more.

  • The authors found that there was a significant difference in EDI-2 total score and body dissatisfaction subscale between the adult and adolescent subgroups. Although there was no significant correlation between those symptom scores and fixation time, it would have been more statistically sound for those scores to be included as a covariate in all analyses involving age group comparisons in order to control for the effects of this factor.

We see the difference in EDI-2 symptoms as a manipulation check that confirms group membership / group differences (AN vs. HC). This difference is therefore also ‘characteristic’ of the respective groups, which is why we would prefer not to include the EDI-2 as a covariate for content-related reasons (in the absence of a significant association with the outcome). From a statistical point of view, there was (as mentioned) no significant correlation, so we would like to keep the procedure as it is. We hope that this is acceptable, thank you!

  • Related to the point already mentioned in the initial main comment, the Results was very hard to follow particularly because multiple tables were often stacked and referred to in the same paragraph. This required a lot of flipping back and forth, looking at the dense tables and then searching for the relevant descriptive text to make sense of the findings. The table labels (S1, S2, etc) were also confusing.

We agree with this impression and apologize, also to the editorial office, for confusing supplement and appendix. We now placed, according to the layout guidelines, supplementary materials in an “appendix” document. Some elements could be removed thanks to leaving out the “time effect” leaving only two tables in the appendix. We hope that, taken together, the paper is “leaner” and clearer structured this way.

  • The discussion about the improvement of body dissatisfaction observed between ages 16-24 found in previous study not being replicated in this study (lines 317-319) seemed out-of-place, as it was not discussed or mentioned anywhere else in the manuscript as a point of interest.

We agree that this discussion point is not necessary and removed it, together with the reference (Lecroix et al.).

  • Could the finding that all participants showed shorter fixation time towards unclothed vs clothed body parts also simply be related to some sort of social desirability effect?

Thank you very much for pointing this out! We added the fact that “directing gaze to clothed rather than unclothed areas may be a strategic, socially guided behavior in the context of cultural or moral expectations” as an alternative explanation for the discussed gaze pattern (ll. 312-314).

  • Lines 415-418 seem to be the only place in the manuscript where the strength or rationale for the study is mentioned, which comes too late in the piece. Related to the main comment, the manuscript lacks a strong rationale from the beginning through to the end about how the study adds to the existing literature and why it is important.

We agree and added some explanatory sentences for the rationale before stating the hypotheses, also for the fact that multiple sets of ROIs were considered in the same images (e.g. ll. 116-117).

  • Various typos were found throughout the manuscript, e.g., line 148 “reactios”, line 156 “dimly-lit” instead of “dimly lighted”, line 291 “und”, line 390 “patters”, etc.

Edited.

Reviewer 2 Report

Comments and Suggestions for Authors

Introduction

In "Therapeutic approaches", including exposure therapies, are one of the therapies that have been shown to be most effective for body dissatisfaction.

Method

Why EDI2 if there is an updated version (EDI3)?

Had participants previously taken a similar assessment, e.g. the silhouette test?

Have the participants undergone therapy or treatment sessions for body dissatisfaction?

Are the images the same for all participants?

Results and Discussion

Would it be possible to include eyetracking measures such as pupil diameter as an emotional measure? 

Unify the format of the graphs (e.g. font).

What could the age differences be due to?

Perhaps another type of graph would be appropriate to express SDs and significance (bar graph?)?

Author Response

  • Dear Reviewer,  Thank you very much for the helpful feedback. We enclose a point-to-point response to your comments.
  •  
  • Introduction
  • In "Therapeutic approaches", including exposure therapies, are one of the therapies that have been shown to be most effective for body dissatisfaction. --> In order to shorten the manuscript and make it more concise, we decided to leave out the effect of two measurements in a small subgroup in the current version of the manuscript. Therefore we also refrained from discussing therapeutic aspects in the introduction and discussion section.
  •  
  • Method
  • Why EDI2 if there is an updated version (EDI3)? --> The EDI-3 is not available as a validated German version yet. For that reason, we used the German version of EDI-2, but updated the citation in the text: Thiel, A., Jacobi, C., Horstmann, S., Paul, T., Nutzinger, D. O., & Schussler, G. (1997, Sep-Oct). [A German version of the Eating Disorder Inventory EDI-2]. Psychother Psychosom Med Psychol, 47(9-10), 365-376. https://www.ncbi.nlm.nih.gov/pubmed/9411465 (Eine deutschsprachige Version des Eating Disorder Inventory EDI-2.)
  • Had participants previously taken a similar assessment, e.g. the silhouette test? --> There has not been conducted a standard assessment, e.g. the silhouette test, when participants took part in the study as this took place at the beginning of inpatient treatment. Some of the participants might have worked with body image therapy material before admission, however, we were not able to control such effects. For future studies it might be interesting to compare different assessment methods of (perceptual) body image like the silhouette test and their correlation with attention patterns.
  • Have the participants undergone therapy or treatment sessions for body dissatisfaction? --> Thank you very much for this valid question. Patients on the ward underwent a CBT therapy program which included body image therapy elements on an individual level. For some of the participants, this included body image exposure therapy, for others/most of them cognitive and behavioral interventions. However, as the treatment program was not completely standardized and did not explicitly involve attentional bias modification, we decided to leave out the part of our research related to treatment effects, as outlined above. 
  • Are the images the same for all participants? --> Yes, all images were the same for all participants. We outlined this in the methods section (ll. 169/170).
  •  
  • Results and Discussion
  • Would it be possible to include eyetracking measures such as pupil diameter as an emotional measure? --> Thank you very much for bringing that issue up. Yes, we are indeed thinking about including pupil diameter in future studies as a measure for emotional reactions towards stimuli. We added a sentence in the discussion section (ll. 359/360).
  • Unify the format of the graphs (e.g. font). --> Edited.
  • What could the age differences be due to? --> As discussed in ll. 338-352 age effects (shorter fixation time within the whole body area in adolescents compared to adults) could be due e.g. to the fact that the photos depicted adult women, who are more appropriate social comparison targets for adults than adolescents (we now elaborated a bit on that with two more recent citations), or the idea that adults might have internalized beauty ideals more strongly. Do you think, a different explanation would be more likely or should we elaborate more on the discussion of the age effects?
  • Perhaps another type of graph would be appropriate to express SDs and significance (bar graph?) --> Thank you very much for the comment. We realized that – probably due to the formatting process – the standard error bars included in prior versions of the graphs were not included in the latest version. We added a reformatted graph including standard error bars and hope that this makes the display of the data clearer.

Reviewer 3 Report

Comments and Suggestions for Authors

Thank you for the opportunity to review this manuscript. I read it with great interest and have provided some suggestions that I hope the authors will fund helpful in revising their manuscript.

  • The abstract is generally well written but I think the authors could make their findings a little clearer to the reader. I think it would be helpful to say from the outset that group differences were not apparent, and then describe the whole sample findings.

Introduction

  • I think the opening sentence of the introduction should be softened a little. Perhaps by using statements like ‘are thought to be characterised…’ to not make the statements too conclusive.
  • Overall, the introduction nicely summarises the literature and is well written. Great work.

Methods

  • Please replace ‘subjects’ with ‘participants’ throughout the manuscript
  • Were eye movements recorded monocularly or binocularly?
  • Clearly presented methods section. Great job, again!

Results

  • I appreciate that BMI percentile is more meaningful or adolescents, but it would be helpful for the reader if you were also to include their BMIs in Table 1 (and would help any other authors who may wish to include your data in a meta-analysis, for example)

Discussion

  • I’d suggest using a more recent reference for the statement regarding comparisons to peers
  • I appreciate that the authors have flagged that the longitudinal results should be interpreted with caution because of the lack of control group, but I think the authors also need to be explicit here that the small sample size is also an important factor in how these results are interpreted
  • It can be difficult to discuss results when hypotheses are not supported – but it’s just as important to have these types of findings published. I commend the authors on their clear methodology, thoughtful interpretations of the results and their balanced discussion of the findings.

Author Response

Dear Reviewer,

We thank you very much for the recognition of our work and the constructive comments. Find enclosed a point-to-point response to the issues you raised: 

  • The abstract is generally well written but I think the authors could make their findings a little clearer to the reader. I think it would be helpful to say from the outset that group differences were not apparent, and then describe the whole sample findings.

Thank you very much for pointing this out; we added a sentence to the abstract to make it clearer (l. 18).

Introduction

  • I think the opening sentence of the introduction should be softened a little. Perhaps by using statements like ‘are thought to be characterised…’ to not make the statements too conclusive.
  • Overall, the introduction nicely summarises the literature and is well written. Great work.

Thank you very much for your appreciation! We edited the opening sentence accordingly (l.30).

Methods

  • Please replace ‘subjects’ with ‘participants’ throughout the manuscript

Thank you for pointing this out; we edited this term throughout the manuscript in order to ensure a more sensitive language use. 

  • Were eye movements recorded monocularly or binocularly?

As stated in l. 159, eye movements were recorded with an infrared video-based binocular tracking system.

  • Clearly presented methods section. Great job, again!

Thank you!

Results

  • I appreciate that BMI percentile is more meaningful or adolescents, but it would be helpful for the reader if you were also to include their BMIs in Table 1 (and would help any other authors who may wish to include your data in a meta-analysis, for example)

We agree and added BMI for the single groups in Table 1.

Discussion

  • I’d suggest using a more recent reference for the statement regarding comparisons to peers

We agree and changed the reference to two more recent ones, focusing on same-aged peers (Burnell et al., 2024) and appearance comparisons (Barbierik et al., 2023).

  • I appreciate that the authors have flagged that the longitudinal results should be interpreted with caution because of the lack of control group, but I think the authors also need to be explicit here that the small sample size is also an important factor in how these results are interpreted

Thank you very much for this constructive remark. Given the indeed very small sample size also observed by the other reviewers, we decided to remove the section on the longitudinal results from the paper.

  • It can be difficult to discuss results when hypotheses are not supported – but it’s just as important to have these types of findings published. I commend the authors on their clear methodology, thoughtful interpretations of the results and their balanced discussion of the findings.

You are very right about not avoiding publishing results contrary to prior hypotheses or earlier research – thank you very much for appreciating this. 

Round 2

Reviewer 2 Report

Comments and Suggestions for Authors

the changes that have been suggested have been incorporated in a gratifying way, providing the necessary clarifications for a better understanding of the methodology and the presentation of the results.

Author Response

Many thanks for your positive reply.